# The Lexicon and Morphosyntax of Child Spanish as Predictors of Inhibition

**DOI:** 10.3390/bs14010012

**Published:** 2023-12-22

**Authors:** John Grinstead, Nina Sorine

**Affiliations:** 1Department of Spanish and Portuguese, The Ohio State University, 1775 College Road, Columbus, OH 43210, USA; 2Department of Hearing, Speech and Language Sciences, Ohio University, Grover Center W218, Athens, OH 45701, USA; ns411716@ohio.edu

**Keywords:** morphosyntax, lexicon, inhibition, Spanish

## Abstract

This study investigates the relationships between lexical development and inhibition, as well as morphosyntax and inhibition, in typically developing monolingual Spanish-speaking children. Recent studies of the relationship between lexical development and inhibition suggest that, as the size of the lexicon increases, so does inhibitory ability. However, the relationship between grammar and inhibition seems more controversial. The work distinguishing the relationships between inhibition and lexicon vs. grammar have been carried out in English, which has relatively impoverished inflectional morphology. Because the relationships considered in the literature are hypothetically not language-particular to English, but rather claims about cognition in general, we would expect to find that they also hold in other languages, including languages with richer morphology, such as Spanish. These considerations led us to ask the following: are expressive and receptive measures of the lexicon and morphosyntax predictive of typically developing monolingual child Spanish-speakers’ inhibitory ability? A sample of 82 monolingual, typically developing Spanish-speaking children in Mexico City were tested with 5 lexical measures, 4 morphosyntax measures, and the Flanker Task measure of inhibition. Results showed that all lexical and morphosyntactic variables correlated significantly with Flanker (*p* < 0.01), except for Number of Different Words (NDW), calculated on the spontaneous production sample. Therefore, inhibition is predicted by lexical development in child Spanish. Additionally, an ever-increasing set of competitor morphological forms requires an ever-increasing inhibitory ability as well.

## 1. Introduction

Executive function can be defined as the way in which behavior is controlled cognitively. Through the use of executive function, an individual can select and monitor their behaviors that facilitate a specific action or actions to achieve a certain goal. Miyake et al. [1] write that there are three important sub-components of executive function, which are inhibition, attention (shifting), and auditory working memory (updating) (see Shah and Miyake (1996) for a plausible distinction between auditory and visual working memory). These abilities are mildly correlated with one another, but also account for unique variance in complex executive function tasks that use more than one component of executive ability [1]. The component that is most important to our study is inhibition, as is the question of how inhibition relates to language development. Inhibition is one’s ability to control impulses or automatic responses through the use of attention and reasoning.

In considering the literature on executive function and language development, we find a prominent stream of work discussing what is referred to as the “bilingual advantage”, perhaps first noted in Peal and Lambert [2]. Though our study does not address the connection between greater inhibitory abilities in bilingual children and their language abilities, this literature nonetheless contains some of the most thorough discussions of the language–executive function connection and is, therefore, reviewed.

In addition to the discussion of executive function and language development in bilingual children, there is also a significant body of literature discussing the connection between language development and executive function in atypical populations. This literature includes some of the best work clarifying the causal relationship between inhibition and language and is, therefore, also discussed.

In the end, our goal is to understand the connection between not only lexical development and inhibition, which seems to be the primary relationship discussed in the literature, but also the relationship between morphosyntax and inhibition. The literature also appears to contain many studies on executive function and either very specific measures of language (e.g., lexical decision tasks) or very gross measures of language (e.g., composite language scores from standardized tests such as the CELF, TOLD, etc.). Our approach is to take a single, widely used measure of inhibition, the Flanker Test [3], and to then compare these results with five distinct measures of the lexicon and with four distinct measures of morphosyntax, in a monolingual Spanish-speaking population in Mexico. Our hope is that in this way, we can learn something about the relationship between inhibition and two distinct domains of language.

## 2. Latent Variables, Structural Equation Modeling, and Miyake’s Model of Executive Function

While there are multiple theories of executive control, we follow the widely accepted model of Miyake et al. [1], which seems empirically very well founded. In Miyake et al. [1], they considered shifting (also called attention), updating (also called auditory working memory), and inhibition as three likely candidate executive functions and examined them at the level of latent variables rather than individually. By studying them as latent variables, Miyake et al. [1] were able to examine what is shared among the multiple exemplary tasks for each executive function, as well as minimize any issue of task impurity. The reason why reducing any problem of task impurity within the study is important is to control variance due to task demands that vary across executive function measures. Miyake et al. [1] explain how previous studies had used correlations and regressions to see how well the participants’ performance was either grouped or separated into certain tasks, including the Wisconsin Card Sort Task (attention), Tower of Hanoi (inhibition), and Operation Span (working memory). The results from these studies, as Miyake et al. [1] described, typically showed a low correlation among the different executive function abilities, which is not usually significant. Additionally, Miyake et al. [1] described how factor analysis within these studies typically separated the results and attributed them to multiple underlying factors.

Miyake et al.’s [1] study consisted of a confirmatory factor analysis to determine, in an adult sample, whether models with one, two, or three latent variables fit better with relatively simple executive function measures. The authors then used the three latent variables that fit best as predictors in a structural equation model that used complex executive function tasks as the outcome variables to determine the degree to which the latent variables explained the complex tasks. To represent each of their different executive functions (attention, auditory working memory, and inhibition), Miyake et al. [1] used 3 different tasks to accompany each executive function, totaling 9 tasks for their 137 participants. The tasks used in Miyake et al.’s [1] study for shifting were Plus–Minus, Number–Letter, and Local–Global. Updating used Keep Track, Tone Monitoring, and Lettering. For inhibition, Miyake et al. [1] used Anti-saccades, Stroop, and Stop–Signal. In addition to these tasks, Miyake et al. [1] included five complex executive tasks as the outcome measures, predicted by the latent variables, which were the Wisconsin Card Sorting Test, Tower of Hanoi, Random Number Generation, Operation Span Task and Dual Tasking. Results from Miyake et al. showed that each latent executive function variable was correlated. From this, Miyake et al. concluded that attention, working memory, and inhibition were separate but related. Although the tasks were independent, they were correlated, and gave an idea of more complex executive function tasks.

Miyake et al. [1] is foundational in the modern study of executive function because it shows that each of the three components can produce variance that loads on single latent variables, using factor analysis. It then shows that each of these latent variables—now independent of specific tasks—can predict components of the variance in complex executive function tasks in a structural equation model. While the latent variable executive function components are somewhat correlated with one another, they are also independent and capable of significantly predicting unique variance in tasks that use more than one component of executive ability.

In other previous literature, executive function and language development have been studied, specifically with children on the autism spectrum and children diagnosed with attention deficit hyperactive disorder. These non-typically developing children are known to have lower executive function (inhibitory) abilities than typically developing children [4,5]. Executive functioning also deals with emotion regulation and controlling impulses, which are said to be lacking in children on the autism spectrum and with children who have attention deficit hyperactive disorder. Furthermore, typically developing bilingual children are argued to have greater inhibitory ability than monolingual children [6,7,8]. Thus, inhibitory executive function ability and language development seem related, but what parts of language are related to them?

In what follows, we first review the substantial literature looking at what Miyake et al. [1] might refer to as “pure” measures of inhibition, paired with different measures of language. Then, we turn to several studies of other aspects of executive function and language development. Finally, we consider the claim by Baayen et al. [9] that inflectional morphology can be stored not only in a stem + affix manner, but also that stem + affix combinations may be stored as complete or wholistic units in the lexicon. This hypothesis is consistent with not only increasing uninflected lexical items predicting greater inhibition, but also with increasing (apparently) inflected lexical items predicting greater inhibition. Next, we turn to an overview of studies that consider the link between inhibition and language development.

## 3. Bilingualism, Language Disorders, Inhibition, and the Lexicon

In the study conducted by Martin-Rhee and Bialystok [6], they specified the degree to which bilingual children show their advantage of being able to perform exceptionally well in specific tasks that require inhibitory control to ignore deceptive perceptual cues. Through Study 1, in which the authors used the Simon task to study inhibiting attention, and Study 2, in which they used the Stroop task to study inhibiting habitual responses, they found that bilinguals were more advanced in their ability to inhibit attention, but that bilinguals and monolinguals were equal when it came to inhibiting a habitual response. The bilingual children who participated in this experiment spoke English, bilingually with French, Chinese, Hebrew, Spanish, or Russian, which the experimenters concluded made no difference in their results. Through Martin-Rhee and Bialystok’s [6] experiment, they were able to provide results that were consistent with Peal and Lambert’s [2] claim that bilinguals are more advanced than monolinguals in their ability to control attention. Inhibiting an incorrect response is vital to controlling attention. Having in-depth knowledge of a language, or, in this case, languages, is crucial to developing stronger inhibitory skills, as we see in our study.

Evidence suggests that bilinguals outperform monolinguals in terms of inhibition. In Blumenfeld and Marian [10], they looked at how processing linguistic ambiguity during auditory comprehension may be associated with inhibitory control. They hypothesized that bilingual experience acts on inhibition mechanisms used during language processing. They compared thirty English-native monolingual speakers and thirty English–Spanish bilingual speakers. The way in which bilinguals were selected was by ensuring that bilinguals had extensive Spanish experience and currently had Spanish exposure. Blumenfeld and Marian [10] administered to them the Language Experience and Proficiency Questionnaire (LEAP-Q) and reported that monolinguals and bilinguals did not differ in terms of English proficiency across comprehension, reading, and speaking. Through trials of Word Recognition/Eye Tracking, as well as Priming Probe trials, the experimenters were able to index activation of competitor words and control words during recognition and to index inhibition of preceding words relative to control words. Participants were asked to identify the quadrant of a visual array containing the target they heard by pressing one of four keys. Immediately following each of the Word Recognition trials, participants were presented with a Priming Probe trial. Participants were then administered the nonlinguistic Stroop task, followed by multiple related linguistic tests. The results from these tasks supported the prediction that, if bilingual experience modulated cognitive control mechanisms associated with language processing, then monolingual and bilingual groups would differ in their use of inhibition to resolve competition between similar-sounding words. Indeed, they concluded that mechanisms working during language comprehension are likely to be influenced by bilingual language experience. One of these mechanisms working during language comprehension is inhibition, which is important for our study.

Blomquist and McMurray [11] investigated the question of how lexicon–internal phonological inhibition might relate to domain-general inhibitory abilities. Specifically, they looked at how we access a target word in our mental lexicon by using an eye-tracking paradigm in school-aged children. While there is evidence that adults display this lexical competition in word recognition with inhibitory connections between words, Blomquist and McMurray [11] used their study to investigate the possible role of inhibition in lexical competition during spoken word recognition in children. They also sought to discover whether this inhibition serves as a mechanism for change in the dynamics of lexical competition across development. Their sample consisted of 46 child participants in 2 different age groups: one a 7–8-year-old age group and the other a 12–13-year-old age group. They employed the Visual World Paradigm (VWP) to investigate lexical inhibition. For each trial, each participant saw four pictures on a computer screen, heard a word, and then selected the picture referent of that word while eye-movements were monitored. Following the VWP task, participants took the spatial Stroop task, to measure inhibition, as well as other subtests. Results suggested that the older children were not able to resolve the lexical interference caused by temporary activation of a competitor as well as the younger children, as efficiency in lexical processing develops in the school-age years, and this development may be linked to changes in underlying competition processes. They concluded that there are clear age-related differences in response to processes in which word recognition occurs. Importantly, they also concluded that there was no significant relationship between inter-lexical inhibition and domain-general inhibition.

Other aspects of executive function have also been investigated for their relation to lexical development, including attention. Dispaldro et al. [12], for example, tested the efficiency of visual engagement in children by measuring their attentional masking. In a sample of 44 Italian children, half of whom had SLI, Dispaldro et al. [12] measured expressive language and receptive language for both the lexicon and grammar present in the study. Their study included measures of the lexicon and grammar, including TVL (Test do Valutazione del Linguaggio), P-IQ (Performance IQ), BNT (Italian version of the Boston Naming Test), an expressive morphosyntax task, PPVT, and TCGB (Test di Comprensione Grammaticale per Bambini). Results showed a significant correlation between attentional masking and the lexicon.

In Kaushanskaya et al. [13], executive function was measured nonverbally. This is helpful in that the linguistic component of these tests does not function as a confounding factor between language and executive function. Working with seventy-one typically developing children, ages eight through eleven, they measured three executive function components, which were inhibition, working memory, and attention, through two nonverbal tasks. Along with this, subjects were also given common standardized language measures. Their results indicated that working memory was significantly associated with the receptive language index on the Clinical Evaluation of Language Fundamentals-Fourth Edition (CELF-4) and nonverbal inhibition was found to be predictive of children’s syntactic abilities. In the study, syntactic abilities are measured by a morphosyntactic grammaticality judgment task, from the TOLD I:4, and from the Concepts and Following Directions subtest of the CELF-4. Thus, at least morphosyntax, in the form of the grammaticality judgment task, appears to be associated with inhibition in English.

## 4. Lexicon, Inhibition, and Causality

It seems clear that there is an association between inhibition and the lexicon in development. But does greater lexical development demand greater inhibitory abilities to manage the larger number of competitors, or does greater inhibitory ability somehow cause the lexicon to become larger? Gangopadhyay et al. [14] used a “cross-lag” design to test the association between lexical processing and inhibition in both English-speaking monolingual children and simultaneous Spanish–English bilingual children. In this design, children were tested once, and then tested again a year later. To measure the lexicon, the authors used an English lexical decision task and, for inhibition, they used two inhibitory tasks: the Flanker task and the Go/No-Go task. Their findings were that later inhibition was predicted by early lexical performance, but later lexical performance was not predicted by early inhibition skills. This was true for both monolingual and bilingual children.

Complicating this picture, Larson et al. [15] partially confirm Gangopadhyay et al.’s [14] findings, again using a cross-lag design. For a sample of typically developing children, they showed that a receptive task of morphosyntax predicted later inhibitory ability, measured by the Flanker task. However, they also showed that, for a sample of children diagnosed with specific language impairment (SLI), early inhibition reaction time predicted later morphological comprehension. This latter finding could appear to contradict the directionality of causality suggested in Gangopadhyay et al.’s [14] study, though it is not entirely clear that reaction time on the Flanker task is an entirely valid measure of inhibitory ability, as it only measures speed and not accuracy.

## 5. Inhibition and Sentence Comprehension

In addition to the previous studies that addressed inhibition (specifically) and language, there are other studies that looked at either more complex executive function tasks or at a range of measures of executive function abilities and language. Minai et al. [16] examined children’s comprehension of universal quantification. They proposed the idea of symmetrical response (SR), which occurs in children in which an atypical semantic interpretation occurs involving a quantifier. The investigated the phenomenon first observed by Philip [17], whereby children, in a picture verification task, are explicitly asked “Is every boy riding an elephant?”, followed by a picture showing some boys each riding an elephant and an extra elephant nobody is riding. In this study, 3–5-year-old children will respond “no” and use the extra elephant as their justification, even though the extra elephant does not falsify the fact that every boy is riding an elephant. Minai et al. [16] reasoned that children reject these sentences by reasoning that the falsifier is the presence of the extra object which ruins the symmetrical one-to-one relation between boys and elephants in the picture. They hypothesized that the extra object, though salient, is irrelevant information that hinders children’s successful universal quantification, which they attributed to children’s still-developing theory of mind (ToM). Based on this hypothesis and previous research, Minai et al. [16] tested a sample of four- and five-year old Japanese-acquiring children in Japan and examined the link between the development of cognitive control and their interpretation of the universal quantifier by using the dimensional change card sort (DCCS) task. The DCCS is a complex executive function task that measures children’s ability to switch perspectives between two competing dimensions that both serve as different standards for card sorting. They used a truth value judgement task (TVJT) to measure children’s interpretations of the universal quantifier. Their results showed that children’s non-adult-like universal quantification with respect to extra-object pictures is considerably affected by their extralinguistic difficulty in switching perspectives using successful cognitive control in picture recognition [16]. They concluded that cognitive control is a factor that influences semantic processing involving universal quantification in children aged four to five.

## 6. Freely Combining vs. Lexically Stored Morphosyntax

Having seen that lexicon and inhibition are associated in development, with less clear results for morphosyntax and inhibition, let us pause to consider the degree to which morphosyntax is lexically stored. In Baayen et al. [9], the issue of the balance of storage and computation for regularly inflected words in Dutch in language comprehension was addressed. One of the arguments that was discussed was one used by, for example, Pinker [18] and Clahsen [19], concerning the occurrence of frequency effects for complex words, which they argued is restricted to irregular complex words. The two token frequency effects, proposed by Baayen [9], are the Surface Frequency Effect and the Base Frequency Effect. The Surface Frequency Effect describes the frequency of complex words, e.g., cant-a-mos (sing–theme vowel–1st Sg. Present Progressive agreement–tense–aspect “We are singing.”). Though the word *cantamos* has three morphemes, the Surface Frequency Effect assumes that the word is stored as a whole. In contrast, the Base Frequency Effect is the product of all the different variants of the root *cant-* being accessed and is taken to be predictive of reaction times that involve accessing this root and its variants (i.e., the lexeme of *cant-*).

The research conducted by Baayen et al. [9] showed very reliable results for Surface Frequency Effects for regular inflected words. Among their experiments, adult Dutch-speakers were asked to judge whether perfect participles were real Dutch words and their answers were timed. Those words that had a high surface frequency were judged significantly faster than were those that were low surface frequency, suggesting that fully inflected Dutch participles could be stored as memorized whole units. According to Baayen [9], this supports the notion that a wide range of linguistic and cognitive factors (e.g., frequency of occurrence, computational complexity, relative costs of storage and computation in mental lexicon) determines the balance of storage and computation. The importance of this study is that it suggests that what appear to be multi-morpheme forms in child language—the kind that show up on standardized tests of morphosyntax—may be stored as whole units in the lexicon. Given what we have seen with increasing lexical scores predicting increasing inhibition scores, Baayen et al.’s [9] results make it seem plausible that we may also find this type of predictive relationship between morphosyntactic development and inhibition.

### Summary

To summarize, language development appears to drive inhibition. When looking at inhibition ability in children, it would make sense that bilingual children have stronger inhibition than monolingual children, as bilingual children have more lexical, and possibly morphosyntactic, competitors for every concept a child wishes to express or comprehend. With more competitors, there theoretically will need to come more inhibitory resources for shutting down the candidate lexical items that are not ultimately correct. In bilinguals, both languages are simultaneously active when processing either language [20,21,22]. The experience of controlling attention between these two languages is a source of practice that boosts those control processes and makes them available for other tasks, such as the perceptual decision tasks used in these experiments [6]. When looking at executive function, a child’s executive function ability is correlated to cognitive abilities like inhibition, which is important to our study. Other measures like fluency and morphosyntax also give us an idea of the cognitive abilities of a child, allowing us to better understand their lexical capabilities, which is important for us in evaluating their inhibition. As we have just seen, it is also possible that morphosyntactic variants of the roots of an inflected language, like Dutch, could be lexically stored, which would logically mean that increasing knowledge of morphosyntax in children should also be predictive of their inhibition abilities. These conclusions lead to the following research questions:Are expressive and receptive measures of the lexicon predictive of typically developing monolingual child Spanish-speakers’ inhibitory ability?;Are expressive and receptive measures of morphosyntax predictive of typically developing monolingual child Spanish-speakers’ inhibitory ability?;If both the lexicon and morphosyntax are predictive of inhibition, do they account for similar proportions of unique variance?

## 7. Methods

### 7.1. Participants

A total of 82 monolingual typically-developing Spanish-speaking children (age range = 50–101 months, mean age = 75.8 months [6 years, three months], SD = 14.7 months) participated in our study. A university IRB-approved consent form was signed for each participant in the study. Participants did not receive compensation for their participation. After childcare institutions were chosen, and approved by a university IRB, families were individually approached at pick-up or drop-off, or were approached in parent meetings, to have the study explained. Parents or guardians either read the consent document or had it read to them by the investigators, in person, before signing. Participants were tested in their preschools, schools, daycare centers, and summer schools in Mexico City.

### 7.2. Procedures

Our lexical and morphosyntactic child language measures were purposefully different test types with different task demands. Some were expressive (e.g., NDW, MLU) and others were receptive (e.g., TVIP, *Comprensión*). The rationale for this was that no single measure is a perfect or “pure” measure of the underlying cognitive construct that it attempts to measure. Rather, the signal produced by all behavioral measures includes some amount of noise in the form of task demands that likely invoke other cognitive or non-cognitive abilities in which the researcher is not interested. In this way, we aspired to capture variance unique to the domains of the lexicon and morphosyntax, if possible.

Children were given five lexical measures: The Peabody Picture Vocabulary Test in Spanish [23]. The Spanish Peabody is a receptive test of the lexicon, normal in Mexico, which asks children to pick one image out of four that corresponds to the word they are given; Number of Different Words (NDW) from a spontaneous speech sample [24]. This is lexical measure that consists of all of the unique words produced by a child during the spontaneous production session. It is calculated automatically by the CLAN programs from the CHILDES Project [25]; NDW from a Frog Story. This is a re-tell protocol, using the *Frog, Where Are You?* book, by Mercer Mayer [26]. Researchers narrate the story of the picture book to children, who are then asked to re-tell the story, lending a narrative character to their language. This contrasts with the more sociolinguistic-interview style spontaneous language sample [27], from which the first NDW calculation was taken; The *Adivinanzas* (“Riddles”) receptive vocabulary subtest of the Batería de Evaluación de Lengua Española (BELE) asks children to guess the word corresponding to a series of clues that they are given. For example, “They are on your face. You use them see with. You close them at night when you go to sleep. What are they?”; Finally, the expressive *Definiciones* (“Definitions”) subtest of the BELE. In this test, children are given a word and asked to define it. Children are given points for non-repetitions of the target word that define the meaning of the word.

For morphosyntax, MLUw or mean length of utterance, calculated in words [28], was calculated from the spontaneous production and Frog Story samples. Previous work has shown that, in Spanish, MLUw correlates 0.9 with MLU calculated in morphemes [29]. Consequently, since the CLAN programs calculate MLUw automatically using our transcriptions, we used MLUw. The receptive *Comprensión* (“Comprehension”) measure from the BELE asks children to choose 1 drawing among 3 that corresponds to a sentence they hear. For example, “Show me where ‘The rabbit is eating the carrot.’” and there are 3 drawings: a girl playing with a doll, a boy playing with a truck, and a rabbit eating a carrot. *Producción Dirigida* (“Elicited Production”) from the BELE is a test in which children are asked to repeat a sentence that corresponds to one of two pictures that the child is shown. For example, a sheet has a drawing of a boy with a truck and another of a rabbit with a carrot. The investigator then instructs the child to repeat what they say and says, “The boy and the truck”. The child’s response is scored on how accurately they repeat the utterance. Children were also given the Flanker Task of inhibition from the EXAMINER Battery [30]. This is a computerized version of the Eriksen and Eriksen [3] Flanker Task, in which children are shown a drawing of 5 fish on a horizontal axis. The fish in the middle occurs above the plus sign. That fish is either oriented in the same direction as the other four, from right to left, or it is oriented in the opposite direction. Children are asked to push the arrow keys on a computer keyboard to indicate the direction (right or left) that the fish is oriented. In order to perform this task when the fish is pointing in the opposite direction of the other fish, the participant must inhibit an instinct to push the arrow key corresponding to most of the fish. An accuracy score, based on the child’s choices in incongruent contexts, is calculated. Reaction time is also calculated. Then, the mean accuracy score is regressed on the child’s mean reaction time, which produces a regression coefficient. This coefficient thus instantiates a measure of the speed–accuracy trade-off and is referred to as the “Flanker Score”.

## 8. Results

### 8.1. Descriptive Statistics

In Table 1, the second row gives the mean scores for our entire sample for each one of our ten measures. In the third row, the standard deviation is given. In the following section, the inferential statistics are given.

### 8.2. Inferential Statistics

Based on the information presented in Table 2, we see that that all lexical variables significantly correlated with inhibition, as measured by the Flanker Task (*p* < 0.01), except for NDW, calculated from the spontaneous production sample (r = 0.212, *p* = 0.055). Additionally, we find that all morphosyntactic variables correlated with Flanker (*p* < 0.01).

To sort out the multicollinearity among our lexical and morphosyntactic variables represented in this table, we performed a stepwise multiple linear regression, with the Flanker Score of Inhibition as our dependent variable. Our predictor variables are our four lexical variables that significantly correlated with Flanker (NDWr, TVIP, *Adivinanzas* and *Definiciones*) and our four morphosyntactic variables that significantly correlated with Flanker (MLUe, MLUr, *Comprensión* and *Producción Dirigida*). In Table 3, we see the two best-fitting models, with all other variables removed from the equations. Variables were removed if the probability of *F* value associated with them was greater than or equal to 0.1, and they were kept in the equation if the probability of *F* was less than or equal to 0.05.

We see that all morphosyntactic variables were eliminated from both best-fitting models and that the two remaining models consist of (1) the Number of Different Words lexical measure, calculated from Frog Stories (NDWr), and (2) the *Adivinanzas* receptive lexical score together with NDWr. The second model has the lower Akaike’s Information Criterion value, which means that it is the preferred model. We can observe in Model 2 that the standardized coefficients of each lexical measure (the third number in the cells) are roughly equivalent, suggesting that expressive NDWr and receptive *Adivinanzas* account for relatively similar proportions of unique variance. Further, the absence of any morphosyntactic variables from the stepwise regression is consistent with their variance being entirely accounted for by the two remaining lexical variables.

## 9. Discussion

Returning to our research questions, we first asked whether increasing lexical development was predictive of inhibition in child Spanish. Based on the present data, we see that inhibition is indeed predicted by lexical development in child Spanish, both receptively and expressively, measured in both controlled and unstructured fashions. For our second research question, we asked whether increasing morphosyntactic development was predictive of inhibition. Again, multiple expressive measures (MLUe, MLUr, Producción Dirigida) and one receptive measure of morphosyntax (*Comprehensión*) were predictive of children’s inhibition scores. The novel findings for morphosyntax are consistent with the hypothesis that an ever-increasing set of competitor morphological forms requires an ever-increasing inhibitory ability. This increasing demand from morphosyntax could arise because of more closed-class morphemes, including inflectional affixes, being added to the lexicon during development. Alternatively, following Baayen et al. [9], Culicover and Nowak [31], and others, children could be adding fully inflected forms, and not just the affixes, to their lexicons. On this view, each of the 47 possible forms of every Spanish verb a child knows could potentially be stored in the lexicon as a morphologically unitary item. Similarly, nouns and adjectives carry inflectional morphology that produce competitor forms that may need to be inhibited. This would obviously require greater lexical storage than stem + affix composition of verb forms.

On the one hand, it seems clear from our morphosyntactic variables and their significant correlations with Flanker that greater morphosyntactic knowledge requires greater inhibitory ability, as does greater lexical knowledge. This finding is consistent with the Baayen type of explanation of lexical storage of inflected forms. Perhaps more strikingly, the lexical component of what our morphosyntactic variables were measuring was apparently so large that our lexical variables accounted for all of their variance, consistent with our multiple regression.

Future work could add in a measure of phrasal syntax to test the degree to which it shows lexical vs. non-lexical properties. This project simply shows that the lexicon and morphosyntax may have similar properties with respect to the part of cognition that has to reduce the number of competitor forms that could correspond to linguistic meaning. We speculate from these results that lexical storage of inflectional morphemes would be just as necessary, if not more so, for domain-general inhibition in languages with more morphology; more lexical storage and more competitors should mean more need for inhibition. Additionally, we speculate from these results that bilingual children who are on the autism spectrum and bilingual children with attention hyperactive deficit disorder would show a greater need for inhibitory control, yet might be at a greater disadvantage than their peers to be able to successfully control certain impulses.

## Figures and Tables

**Table 1 behavsci-14-00012-t001:** Means and standard deviations of our measures of the lexicon, morphosyntax, and inhibition.

	TVIP	NDWe	NDWr	Adiv.	Def.	Compren.	Prod. Dir.	MLUwe	MLUwr	Flanker
Mean	67.90	359.12	109.19	16.55	42.52	31.94	41.74	4.75	5.80	5.64
SD	17.15	82.54	20.95	5.49	11.87	5.59	9.15	0.97	0.86	1.66

**Table 2 behavsci-14-00012-t002:** Pearson product moment correlations of inhibitory, lexical and morphosyntactic measures—Note: *** p <* 0.01.

		Inhibition	Lexicon					Morphosyntax			
		Flanker	NDWe	NDWr	TVIP	Adiv	Def	MLUe	MLUr	Comp	Prod
Inhib	Flanker		0.212	0.499 **	0.452**	0.453 **	0.394 **	0.287 **	0.357 **	0.468 **	0.458 **
Lex	NDWe			0.485 **	0.452**	0.327 **	0.400 **	0.606 **	0.411 **	0.396 **	0.366 **
	NDWr				0.475 **	0.389 **	0.429 **	0.445 **	0.627 **	0.432 **	0.546 **
	TVIP					0.643 **	0.694 **	0.508 **	0.461 **	0.617 **	0.651 **
	*Adivinanzas*						0.516 **	0.443 **	0.316 **	0.499 **	0.626 **
	*Definiciones*							0.511 **	0.476 **	0.458 **	0.557 **
MorSyn	MLUe								0.482 **	0.427 **	0.421 **
	MLUr									0.341 **	0.546 **
	*Comprehensión*										0.531 **
	*Produción Dirigida*										

**Table 3 behavsci-14-00012-t003:** Multiple regression of 4 lexical measures and 4 morphosyntactic variables on the Flanker Test of Inhibition. Coefficients, (standard errors), and standardized coefficients. Note: * *p* < 0.05, *** *p* < 0.001.

Predictors	Null	Model 1	Model 2
NDWr		0.032 ***	0.023 *
		(0.008)	(0.009)
		0.392	0.279
*Adivinanzas*			0.087 *
			(0.033)
			0.284
Constant	5.594 ***	2.154 *	1.716
AIC	324.88	303.51	298.705

## Data Availability

The data presented in this study are available upon request from the corresponding author. The data are not publicly available because of the restrictions set by the ethics committee review.

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
