# Peer review of "The Lexicon and Morphosyntax of Child Spanish as Predictors of Inhibition"

_behavsci, 2023, doi:10.3390/bs14010012_

Round 1

Reviewer 1 Report

Comments and Suggestions for Authors

Line 93: correct "Miyake et al’s" to "Miyake et al.’s"

Lines 348-9: headings 7 and 7.1 are orphan lines; Move to next page.

Line 417: Novak appears in line 473 as Nowak; correct as appropriate; also in line 473, the date (2008) is not enclosed in parentheses.

Author Response

Dear Reviewer,

Thank you for the corrections. We have made them all.

Sincerely,

John Grinstead

Reviewer 2 Report

Comments and Suggestions for Authors

I think the authors can produce a much better manuscript.  The data are potentially interesting.  Here are some suggestions for how they could proceed.

1  1. Include all and only relevant material in the introduction.  First, there is a lot of irrelevant information.  For example, it is not clear why the (disputed) data on cognitive decline are included.  Second, the authors could include more recent data on everything from executive function to inconsistent results on the relation between language and executive function.  With respect to executive function, the authors do not cite Miyake, A., & Friedman, N. P. (2012). The nature and organization of individual differences in executive functions: Four general conclusions. Current Directions in Psychological Science21(1), 8-14.  That is an update of the earlier paper they do cite.  The discussion of executive function in the manuscript is too long and diffuse.  With respect to inconsistencies, there have now been a number of narrative reviews and meta-analyses casting doubt on effects of bilingualism on cognition.

22.   Since the focus is on vocabulary and morphosyntax, some theoretical background indicating why those skills would be related to inhibition would be helpful.  I did not see a focused argument for expecting language to improve inhibition in monolingual children.  The authors might have said, for example, that having a larger vocabulary requires more exercise of executive control.

33.  The method and analyses are under-described and under-motivated.  The authors do not address the issue that all of their measures (but one) are moderately to highly correlated with each other.  Maybe children who are good at x are also good at y.  That does not imply any causal relationship.  I do not understand why the authors only computed correlations.  I also do not understand their flanker measure.

44.  The authors have very little to say about the import of their findings.

Small points

11.  Most studies suggest better performance on the Stroop with age, not worse performance.

22.   I do not think that the word “lexicon” can appear without the word “the” or some other determiner in front of it.

Comments on the Quality of English Language

The language is basically fine, except for the odd use of "lexicon".

Author Response

Response to Reviewer 2

1 1. Include all and only relevant material in the introduction. First, there is a lot of irrelevant information. For example, it is not clear why the (disputed) data on cognitive decline are included. Second, the authors could include more recent data on everything from executive function to inconsistent results on the relation between language and executive function. With respect to executive function, the authors do not cite Miyake, A., & Friedman, N. P. (2012). The nature and organization of individual differences in executive functions: Four general conclusions. Current Directions in Psychological Science, 21(1), 8-14. That is an update of the earlier paper they do cite. The discussion of executive function in the manuscript is too long and diffuse. With respect to inconsistencies, there have now been a number of narrative reviews and meta- analyses casting doubt on effects of bilingualism on cognition.

We thank the reviewer for their feedback. We have deleted the reference to Bialystok and Craik (2010). Indeed, while the conclusions of the study relate executive function (EF) and language, we have a pretty busy literature review and if anything is going to go, this reference to cognitive decline and language is a good candidate.

Consistent with the reviewer’s suggestion, we would not want to include Miyake and Friedman (2012), since it would not add anything to the view of EF as it relates to language. To wit, there is nothing outside of the “Unity and Diversity” section that relates to our EF-language argument: the genetic dimensions do not, clinical and societal relevance do not and developmental stability, measured among adolescents, does not.

This leaves:

  • Blumenfeld and Marian (2011) – directly relevant to language and EF
  • Blomquist and McMurray (2017) – directly relevant to lexicon and EF
  • Dispaldro et al. (2013) – directly relevant to lexicon and EF
  • Kaushanskaya et al. (2017) – directly relevant to language and EF
  • Gangopadhyay et al. (2019) – directly relevant to lexicon and EF
  • Larson et al. (2020) – directly relevant to morphosyntax and EF
  • Minai et al. (2012) – directly relevant to language and EF

We do not believe that any of these references could responsibly be left out of the review, given their importance for what we want to argue. Meta-analyses narrative reviews of EF, on the other hand, which do not relate to language, would indeed be superfluous.

Our position that our literature review is appropriate is substantiated by Reviewer 3, who says, “Very clear and detailed account of a wide range of studies on executive function, inhibition, as well as on the language skills of monolingual and bilingual individuals. I think the very comprehensive review of these studies really help build the rationale for this current study.”

  1. Since the focus is on vocabulary and morphosyntax, some theoretical background indicating why those skills would be related to inhibition would be helpful. I did not see a focused argument for expecting language to improve inhibition in monolingual children. The authors might have said, for example, that having a larger vocabulary requires more exercise of executive control.

We directly address this concern in section 4 of the manuscript, entitled “Lexicon, Inhibition and Causality” In the first paragraph we say:

It seems clear that there is an association between inhibition and lexicon in development. But, does greater lexical development demand greater inhibitory abilities to manage the larger number of competitors, or does greater inhibitory ability somehow cause the lexicon to become larger? Gangopadhyay et al. (2019) used a “cross-lag” design to test the association between lexical processing and inhibition in both English-speaking monolingual children and simultaneous Spanish-English bilingual children. In this design, children are tested once, and then tested again a year later. To measure lexicon, the authors used an English lexical decision task and for inhibition, they used two inhibitory tasks: the Flanker task and the Go/No Go task. Their findings were that later inhibition was predicted by early lexical performance, but later lexical performance was not predicted by early inhibition skills. This was true for both monolingual and bilingual children.

This is a causal theory of developing lexicon causing growth in inhibition.

Similarly, in our “Summary” section, we say:

Summary

To summarize, language development appears to drive inhibition. When looking at inhibition ability in children, it would make sense that bilingual children have stronger inhibition than monolingual children do, as bilingual children have a more expansive knowledge of language. Both languages are simultaneously active when a bilingual is using one of them (e.g., Grainger and Beauvillain, 1987; Brysbaert, 1998; Kroll and Dijkstra, 2002). The experience of controlling attention between these two languages is a source of practice that boosts those control processes and makes them available for other tasks, such as the perceptual decision tasks used in these experiments. (Martin-Rhee & Bialystok, 2008, pp. 91-92). When looking at executive function, a child’s executive function ability is correlated to cognitive abilities like inhibition, which is important to our study. Other measures like fluency and morphosyntax also give us an idea of the cognitive abilities of a child, allowing us to better understand their lexical capabilities, which is important for us in evaluating their inhibition. As we have just seen, it is also possible that morphosyntactic variants of the roots of an inflected language, like Dutch, could be lexically stored, which would logically mean that increasing knowledge of morphosyntax in children should also be predictive of their inhibition abilities.

Again, the conclusion that we are hypothesizing that a larger vocabulary requires more exercise of executive control seems to be stated clearly here. We repeat it in the Conclusion.

  1. The method and analyses are under-described and under- motivated. The authors do not address the issue that all of their measures (but one) are moderately to highly correlated with each other. Maybe children who are good at x are also good at y. That does not imply any causal relationship. I do not understand why the authors only computed correlations. I also do not understand their flanker measure.

We thank the reviewer for pointing this out. Readers may not be familiar with our measures. We have now included descriptions of them, with references.

A Pearson Product Moment Correlation and a linear regression coefficient are identical mathematical values. Bivariate correlations are easy to understand, so we used them. Nothing theoretical was riding on which of these measures was more predictive than the other, so there would have been no purpose served by a multiple linear regression.

  1. The authors have very little to say about the import of their findings.

Small Points

  1. Most studies suggest better performance on the Stroop with age, not worse performance.

We are not sure what the reviewer is referring to here. We gave the Flanker, not the Stroop, test of inhibition.

  1. I don’t think that the word “lexicon” can appear without the word “the” or some other determiner in front of it.

We have systematically changed all of the relevant cases of “lexicon” to “the lexicon”.

Reviewer 3 Report

Comments and Suggestions for Authors

Summary

This manuscript reports results from an experimental study with monolingual Spanish-speaking children investigating the relationship between lexical development and inhibition.

Thank you for the opportunity to review this manuscript. I enjoyed reading this manuscript and I think it has considerable potential to contribute to the literature around language processing in several ways.

There seems to be an imbalance in this manuscript in my opinion between the background literature sections, and the results and discussion. There seems to be a much larger focus (and word count) on this manuscript on the literature review instead of the results and discussion, which should actually be the more detailed and meatier sections.

Introduction (and background literature sections)

Very clear and detailed account of a wide range of studies on executive function, inhibition, as well as on the language skills of monolingual and bilingual individuals. I think the very comprehensive review of these studies really help build the rationale for this current study.

I think the details about the current study would potentially be positioned better after the literature review details instead of after the Introduction.

Method

In the participants section, please include further details about how the participants for this study were recruited. Were there any incentives or compensation?

It is also important in my opinion to include some further details about these tasks in the same way that there is description for the reading tasks, so that it is clear what participants were required to do in each task.

The procedure section also requires much further elaboration around the data collection processes. Were participants invited to the university lab or were the tests administered at the school settings? These details are essential as part of the method followed as part of this study.

I would also appreciate some more details around the rationale for selecting these five lexical measures in this section.

Results

This section is quite brief. Further details that unpack the results presented in the table are needed. I understand that the focus is on the correlations, but further details are needed to explain for the reader how these significant correlations between measures are important for this study.

Discussion

This section is also quite brief. The details included are relevant, but I believe that more details are needed around the individual results, as well as on how the specific results relate to previous literature. The importance and relevance of the results need to be presented in much more detail in this section. This would really help the reader better understand the authors’ interpretation of the results and how these relate to the literature presented.

Referencing

Citations need to be presented in numbers within brackets (e.g. [1]), which did not seem to be the case in this manuscript. I would therefore recommend significant changes to the way that in-text citations and the reference list are presented in order to be in line with MDPI’s guidelines.

Author Response

Response to Reviewer 3

This manuscript reports results from an experimental study with monolingual Spanish-speaking children investigating the relationship between lexical development and inhibition.

Thank you for the opportunity to review this manuscript. I enjoyed reading this manuscript and I think it has considerable potential to contribute to the literature around language processing in several ways.

There seems to be an imbalance in this manuscript in my opinion between the background literature sections, and the results and discussion. There seems to be a much larger focus (and word count) on this manuscript on the literature review instead of the results and discussion, which should actually be the more detailed and meatier sections.

We thank the reviewer for this feedback. We have substantially increased our description of our measures in the Procedures section. Further, we have added a multiple regression, which shows that the variance accounted for by our morphological variables and their significant correlations with inhibition is entirely accounted for by our lexical variables. We have added discussion of this in our Discussion section.

Introduction (and background literature sections)

Very clear and detailed account of a wide range of studies on executive function, inhibition, as well as on the language skills of monolingual and bilingual individuals. I think the very comprehensive review of these studies really help build the rationale for this current study.

I think the details about the current study would potentially be positioned better after the literature review details instead of after the Introduction.

Forgive us, but we think that the details of our study, that is, our Procedures section does currently come after the literature review. Sorry if we have failed to follow the comment.

Method

In the participants section, please include further details about how the participants for this study were recruited. Were there any incentives or compensation?

There were no incentives or compensation. We now say this in our Participants section.

It is also important in my opinion to include some further details about these tasks in the same way that there is description for the reading tasks, so that it is clear what participants were required to do in each task.

We have now included descriptions of the tasks in the Procedures section.

The procedure section also requires much further elaboration around the data collection processes. Were participants invited to the university lab or were the tests administered at the school settings? These details are essential as part of the method followed as part of this study.

We now say where children were tested.

I would also appreciate some more details around the rationale for selecting these five lexical measures in this section.

They are different kinds of lexical measures. The rationale was that no measure is a “pure” measure of the underlying construct being tested. They all have some sort of task demands and consequently have some overlapping variance, but not total overlapping variance. This is hopefully somewhat evident by the fact that some are receptive and some are expressive. We now say this.

Results

This section is quite brief. Further details that unpack the results presented in the table are needed. I understand that the focus is on the correlations, but further details are needed to explain for the reader how these significant correlations between measures are important for this study.

We have added greater detail to our explanation, along with an additional statistical test. We are also adhering to the APA restriction on describing anything other than the statistical properties and findings in this section, leaving a discussion of their meaning to the Discussion section.

Discussion

This section is also quite brief. The details included are relevant, but I believe that more details are needed around the individual results, as well as on how the specific results relate to previous literature. The importance and relevance of the results need to be presented in much more detail in this section. This would really help the reader better understand the authors’ interpretation of the results and how these relate to the literature presented.

We have now attempted to expand the Discussion section.

Referencing

Citations need to be presented in numbers within brackets (e.g. [1]), which did not seem to be the case in this manuscript. I would therefore recommend significant changes to the way that in-text citations and the reference list are presented in order to be in line with MDPI’s guidelines.

We have now changed our references to match MDPI’s guidelines.

Round 2

Reviewer 3 Report

Comments and Suggestions for Authors

I would like to thank the authors for their hard work in revising this manuscript. I really appreciate the detailed responses to my comments on the first version and I am happy with all the changes that have been made. I have just noted a few more minor changes below:

·         In the Participants section, I think some more details are still needed around how the participants were recruited (did you find lists of educational settings on a government database and randomly selected some to contact? Did you have your own contacts?) Mainly around the process through which these educational settings were recruited, as well as the details of the process of how the participants were recruited (did the parents receive a letter to consent for their child’s participation? Was there a flyer at the settings asking interested parents to express their interest via email?)

I am looking forward to reading the final version once it is published.

Author Response

Dear Reviewer 3,

Thank you for your feedback, and kind words.

Please find our response below.

best,

John and Nina

I would like to thank the authors for their hard work in revising this manuscript. I really appreciate the detailed responses to my comments on the first version and I am happy with all the changes that have been made. I have just noted a few more minor changes below:

  • In the Participants section, I think some more details are still needed around how the participants were recruited (did you find lists of educational settings on a government database and randomly selected some to contact? Did you have your own contacts?) Mainly around the process through which these educational settings were recruited, as well as the details of the process of how the participants were recruited (did the parents receive a letter to consent for their child’s participation? Was there a flyer at the settings asking interested parents to express their interest via email?)

I am looking forward to reading the final version once it is published.

In response to the reviewer's comment, we have added additional details to the Participants section. Frankly, we chose data collection sites that were easily navigable by our mostly working-class RAs, who did not have cars, and needed to be able to get around Mexico City on the subway and in peseros (a type of bus system in Mexico City). Because this fact has no bearing on the age, class, gender or other dimensions of our sample's linguistic and cognitive characteristics, but rather is one of a thousand micro-factors deciding how we conducted the study, we have left it out.
